# Short-term treatment with a peroxisome proliferator-activated receptor α agonist influences plasma one-carbon metabolites and B-vitamin status in rats

**Vegard Lysne**[1]*, **Bodil Bjørndal**[2], **Mari Lausund Grinna**[2], **Øivind Midttun**[3], **Per Magne Ueland**[2,3], **Rolf Kristian Berge**[2,4], **Jutta Dierkes**[5,6,7], **Ottar Nygård**[1,4,6], **Elin Strand**[2]

**1** Centre for Nutrition, Department of Clinical Science, University of Bergen, Bergen, Norway, **2** Department of Clinical Science, University of Bergen, Bergen, Norway, **3** Bevital A/S, Bergen, Norway, **4** Department of Heart Disease, Haukeland University Hospital, Bergen, Norway, **5** Centre for Nutrition, Department of Clinical Medicine, University of Bergen, Bergen, Norway, **6** Mohn Nutrition Research Laboratory, Centre for Nutrition, University of Bergen, Bergen, Norway, **7** Laboratory Medicine and Pathology, Haukeland University Hospital, Bergen, Norway

* vegard.lysne@uib.no

**Data Availability Statement:** All relevant data are within the manuscript and its Supporting Information files.

## Abstract

### Introduction

Peroxisome proliferator-activated receptors (PPARs) have been suggested to be involved in the regulation of one-carbon metabolism. Previously we have reported effects on plasma concentrations of metabolites along these pathways as well as markers of B-vitamin status in rats following treatment with a pan-PPAR agonist. Here we aimed to investigate the effect on these metabolites after specific activation of the PPARα and PPARγ subtypes.

### Methods

For a period of 12 days, Male Wistar rats (n = 20) were randomly allocated to receive treatment with the PPARα agonist WY-14.643 (n = 6), the PPARγ agonist rosiglitazone (n = 6) or placebo (n = 8). The animals were sacrificed under fasting conditions, and plasma concentration of metabolites were determined. Group differences were assessed by one-way ANOVA, and planned comparisons were performed for both active treatment groups towards the control group.

### Results

Treatment with a PPARα agonist was associated with increased plasma concentrations of most biomarkers, with the most pronounced differences observed for betaine, dimethylglycine, glycine, nicotinamide, methylnicotinamide, pyridoxal and methylmalonic acid. Lower levels were observed for flavin mononucleotide. Fewer associations were observed after treatment with a PPARγ agonist, and the most notable was increased plasma serine.

**Funding:** Authors ØM and PMU are employed by BEVITAL A/S (www.bevital.no), who performed the biochemical analyses for this study. The funder provided support in the form of salaries for authors [ØM and PMU], but did not have any additional role in the study design, data collection and analysis, decision to publish, or preparation of the manuscript. The specific roles of these authors are articulated in the 'author contributions' section.

**Competing interests:** I have read the journal's policy and the authors of this manuscript have the following competing interests: ØM and PMU are employed by BEVITAL A/S (www.bevital.no), who performed the biochemical analyses for this study. This commercial affiliation does not alter our adherence to PLOS ONE policies on sharing data and materials. All other authors have declared no competing interests.

**Abbreviations:** ACMS, Aminocarboxymuconate semialdehyde; BCAA, Branched-chain amino acids; BHMT, Betaine-homocysteine methyltransferase; DMG, Dimethylglycine; DMGDH, Dimethylglycine dehydrogenase; FMN, Flavin mononucleotide; GNMT, Glycine-N-methyltransferase; gSD, Geometric standard deviation; GSH, Glutathione; MMA, Methylmalonic acid; mNAM, $N^1$-methylnicotinamide; MS, Methinine synthase; mTHF, 5'-methyltetrahydrofolate; NA, Nicotinic acid; NAM, Nicotinamide; PA, Pyridoxic acid; PAr, PAr-index; PL, Pyridoxal; PLP, Pyridoxal-5'-phosphate; PPAR, Peroxisome proliferator-activated receptor; QAPRT, Quinolinic acid phosphoribosyltransferase; SARDH, Sarcosine dehydrogenase; SMD, Standardized mean difference; TTA, Tetradecylthioacetic acid.

## Conclusion

Treatment with a PPARα agonist influenced plasma concentration of one-carbon metabolites and markers of B-vitamin status. This confirms previous findings, suggesting specific involvement of PPARα in the regulation of these metabolic pathways as well as the status of closely related B-vitamins.

## Introduction

The choline oxidation pathway consists of the metabolic reactions converting choline, originating from endogenous synthesis or from the diet, to glycine via the intermediate metabolites betaine, dimethylglycine (DMG) and sarcosine. Glycine may also be reversibly synthesized from serine via serine-hydroxymethyltransferase. The choline oxidation pathway is closely related to the homocysteine-methionine cycle, through the enzyme betaine-homocysteine methyl transferase (BHMT) [1]. In this reaction a methyl group is transferred from betaine to homocysteine, producing methionine and DMG. The other homocysteine remethylation pathway is catalyzed by methionine synthase (MS), where the methyl group is transferred from 5'-methyltetrahydrofolate (mTHF) via the MS bound cofactor methylcobalamin (vitamin B12) forming methionine. Alternatively to remethylation, homocysteine may be irreversibly catabolized through the transsulfuration pathway, by the enzymes cystathionine-β-synthase and cystathionine-γ-lyase (Fig 1) [2]. Plasma and urinary concentration of metabolites in this pathway, including choline [3–5], betaine [5–7], DMG [8–10], sarcosine [11, 12], glycine and serine [13–15], have been linked to risk of major lifestyle diseases such as diabetes, cancer and cardiovascular disease.

Peroxisome proliferator-activated receptors (PPARs) are a family of nuclear receptors involved in the regulation of a variety of metabolic functions, including different aspects of energy metabolism [16, 17]. Recent studies in animals have implicated PPARs, in particular the PPARα subtype, in the regulation of one-carbon metabolism pathways. Down regulation of the genes encoding DMG dehydrogenase (DMGDH), sarcosine dehydrogenase (SARDH) and glycine-N-methyltransferase (GNMT) of the choline oxidation pathway [18], as well as both enzymes of the transsulfuration pathway [18, 19], have been observed after PPARα activation in rats. Down regulation on the protein level of BHMT, DMGDH, SARDH and GNMT have also been observed in rats and mice [20, 21].

Treatment of humans with fibrates, which are PPARα agonists, have consistently been associated with increased plasma total homocysteine (tHcy) [22, 23], and also decreased plasma betaine concentrations [24, 25] and increased urinary output of choline, betaine and DMG [24–27]. In animals, PPARα activation has been associated with increased plasma DMG [28] as well as increased glycine and serine [18, 28–31]. PPARα activation in animals have also been linked to altered status of several B-vitamins, such as increased plasma concentrations of riboflavin [31], nicotinamide (NAM) and $N^1$-methylnicotinamide (mNAM) [28, 31], the vitamin B6 indices pyridoxal (PL) and pyridoxal-5'-phosphate (PLP) [28] and the functional marker of cobalamin deficiency, methylmalonic acid (MMA) [28], and reduced plasma folate [28]. Reduced folate has also been reported in humans by some [32], but not all [33]. Increased urinary output of NAM and mNAM have been consistently reported in rodents [18, 28, 31, 34, 35].

In a previous study of male Wistar rats [28], we demonstrated that long-term treatment with the pan-PPAR agonist tetradecylthioacetic acid (TTA) was associated with alteration in

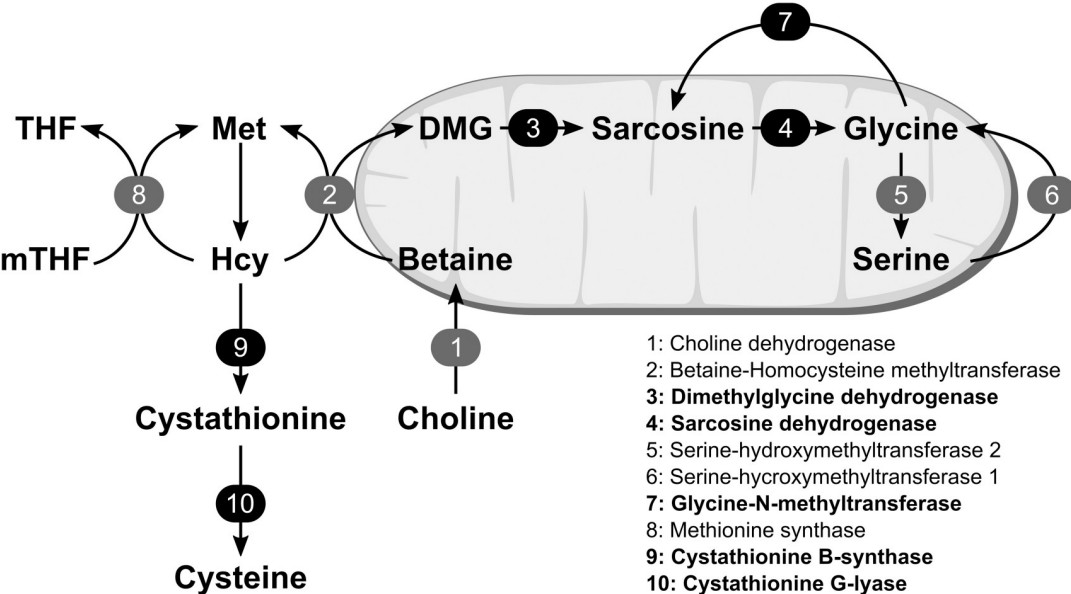

**Fig 1. Overview of metabolic pathways discussed.** The numbers indicate enzymes, and the black boxes indicate enzymes of which gene expression have been previously shown to be down regulated by PPARα activation. The figure indicates where the enzymatic reactions occur (within/outside mitocondria). DMG indicates dimethylglycine; Hcy, homocysteine; Met, methionine; mTHF, 5'-methyltetrahydrofolate and THF, tetrahydrofolate.

the plasma concentration of several metabolites along the choline oxidation pathway as well as markers of B-vitamin status. However, as TTA is a pan-PPAR agonist we could not conclude whether this was a result of PPARα-activation, activation of other PPAR subtypes or PPAR-independent effects of the TTA treatment.

In the current study, we aim to extend on our previous findings and clarify the role of PPAR-activation on plasma concentration of one-carbon metabolites, as well as the status of related B-vitamins. We measured these metabolites in plasma from animals treated with specific PPARα and PPARγ agonists.

## Materials and methods

### Animals and study design

The current study is a substudy of an experiment aiming to explore the role of short-term activation of PPARs on lipid and fatty acid composition in liver and heart, and the experimental setup has been described in detail elsewhere [36].

Briefly, a total of 20 male Wistar rats (Taconic Europe A/S, Ry, Denmark), aged 8 weeks and with a body weight of 200-225g, were randomized into three groups: 1) Control (n = 8), 2) PPARα agonist (PPARα, n = 6); and 3) PPARγ agonist (PPARγ, n = 6). During 12 days of intervention, the animals received standard low-fat chow diet, and the daily treatments were given as 300 μl of muffin dough as vehicle, containing the active agent or placebo. The control group received pure vehicle, the PPARα group received 20 mg/kg/day of WY-14.643 (Tocris Bioscience, Bristol, UK) and the PPARγ group received 10 mg/kg/day Rosiglitazone (Sigma-Aldrich, St. Louis, MO). The muffin dough vehicle contained eggs, sugar, gluten free flour, vanilla sugar, milk and butter.

The animals were housed 2–3 per cage, and had free access to chow diet and tap water during the full study period. Prior to the study, the animals were habituated to the cage conditions

and experimental handling, and introduced to the muffin dough vehicle. Animals in the same cage belonged to the same experimental group, but were taken out of the cages to receive treatment to ensure equal dosing. Cage placement, sequence of animal handling and termination were block randomized. After 12 days, the animals were sacrificed under fasting conditions, by cardiac puncture and exsanguination after receiving anesthesia with 2% isoflurane (Schering-Plough, Kent, UK).

Feed intake was estimated by weighing the food provided to the cages, as well as residual feed. During the intervention, feed intake and total weight gain was somewhat higher in animals treated with PPARγ-agonist, but liver weight was increased in rats treated with PPARα-agonist. In the PPARα-group, hepatic expression of *PPARα* was 61% higher, and the expression of known PPARα target genes such as *Acox1* (9-fold increase), *Cd36* (9-fold increase), *LPL* (6-fold increase) and *Hmgcs2* (2-fold increase) were affected by treatment with PPARα-agonist, demonstrating activation of PPARα in the liver [36]. Further, adipose tissue expression of known PPARγ target genes, such as Fatp1 (3-fold increase) and Fabp4 (0.5-fold increase) were affected by treatment with PPARγ agonist, demonstrating activation of PPARγ [36].

### Ethics statement

The animal experiments complied with the Guidelines for the Care and Use of Experimental Animal use and the study protocols were approved by the national animal research authority (FOTS, ID number 2014/6187).

### Biochemical analyses

Quantification of plasma metabolites were performed at Bevital A/S (Bevital, Bergen, Norway, www.bevital.no), using gas- or liquid chromatography coupled with tandem mass spectrometry for all metabolites except cobalamin which was analyzed by microbiological assay [37–40]. For the purpose of this targeted metabolomics approach, the metabolites of interest included the metabolites of the methionine-homocysteine cycle and the transsulfuration pathway (methionine, tHcy, cystathionine and cysteine), the choline oxidation pathway (choline, betaine, DMG, glycine and serine), as well as markers of related B-vitamins (Flavin mononucleotide [FMN], NAM, mNAM, Nicotinic acid [NA], PL, PLP, 4-pyridoxic acid (PA), the PAr-index (Par, calculated as the ratio of 4-pyridoxic acid divided by the sum of pyridoxal 5'-phosphate plus pyridoxal), mTHF, cobalamin and MMA).

### Statistical analyses and presentation of data

The animals were housed 2–3 per cage, but as the rats were taken out of the cages to receive the intervention, the individual rat was regarded as the experimental unit of analysis. Plasma metabolite concentrations were log-transformed, and the data are presented as geometric means (geometric standard deviation, gSD). The groups were compared by one-way ANOVA, and the proportion of variance explained by the experimental groups were assessed by calculating the $\eta^2$. The assumption of equal variances was assessed with Levene's test and visually by plotting the residuals. Within-group normality was visualized by Q-Q plots of the residuals. Planned comparisons towards the control group were performed for the two intervention groups. Standardized mean difference (SMD; 95% confidence interval) were calculated and plotted to illustrate the differences from the control group. For effect sizes, Cohen's cutoff of $\eta^2$ > 35% was considered a large proportion of explained variance [41], and SMD > 1 were considered a large effect size. All individual data points are presented in S1 Fig. To evaluate

potential cage effects, data was also plotted treating the cages as the experimental unit (n = 2–3 per group, S2 Fig).

As no new experiments were performed for this investigation, no formal power calculation was conducted. However, the current sample size was considered sufficient for replication of the most pronounces differences previously reported (SMD > 3) [28]. At a conventional cutoff for statistical significance at p < 0.05 the current sample sizes would yield 71% power to detect a large variance explained ($\eta^2 > 35\%$), and 80% power of detecting between-group differences of SMD > 1.65.

Statistical analyses were performed using R software version 3.5.1 [42], and the packages within the *tidyverse* (*dplyr*, *broom*, *purrr*, *magrittr*, *rlang) and forestplot*.

## Results

The plasma concentration of metabolites and the SMD (95% CI) for the intervention groups versus the control group are given in **Fig 2**. Overall, more and larger differences were observed after treatment with PPARα agonist compared to PPARγ agonist.

Methionine ($\eta^2$ = 52%, p = 0.009), homocysteine ($\eta^2$ = 48%, p = 0.016) and cystathionine ($\eta^2$ = 39%, p = 0.048) were different between the groups. Planned comparisons showed that methionine (SMD = -1.5), homocysteine (SMD = -1.4) and cystathionine (SMD = -0.88) were lower after treatment with PPARγ agonist, while all metabolites were higher after treatment with PPARα agonist(SMD = 1.0, 1.0 and 0.9, respectively).

In the choline oxidation pathway, group differences were observed for betaine ($\eta^2$ = 68%, p < 0.001), DMG ($\eta^2$ = 68%, p < 0.001), glycine ($\eta^2$ = 81%, p < 0.001), and serine ($\eta^2$ = 60%, p < 0.001). The planned comparisons demonstrated higher concentrations of betaine (SMD = 2.7), DMG (SMD = 3.4), glycine (SMD = 4.4) and serine (SMD = 2.4) in the PPARα group. Lower plasma choline (SMD = -1.5), and higher serine (d = 1.9) was observed in the PPARγ group.

FMN, but not riboflavin, was different between the groups ($\eta^2$ = 65%, p < 0.001), and the planned comparisons revealed this to be due to lower concentrations in the PPARα group (SMD = -2.4). NAM and mNAM were different between groups (both $\eta^2 > 90\%$, p < 0.001), and this was entirely explained by higher levels after treatment with PPARα agonist (SMD = 6.6 and 5.9, respectively). The same pattern was seen for the two vitamin B6 forms PL and PLP ($\eta^2$ = 74% and 48%, both p < 0.001), where higher concentrations in the PPARα agonist group explained the difference (SMD = 3.9 and 2.0, respectively). A between-group difference was observed for cobalamin ($\eta^2$ = 43%, p = 0.03) and its functional marker MMA ($\eta^2$ = 73%, p < 0.001). PPARα agonistic treatment was associated with lower plasma cobalamin (SMD = -1.8) and higher MMA (SMD = 3.0).

## Discussion

This short-term study of one-carbon metabolites and B-vitamins in male Wistar rats treated with PPAR agonists demonstrated pronounced effects of PPARα agonist on the plasma concentration of metabolites of the choline oxidation pathway and markers of B-vitamin status. The largest differences were seen for plasma betaine, DMG, glycine, NAM, mNAM, PL and MMA, which were all higher, and FMN which was lower in the PPARα group. The current findings confirm our previous observations on effects of the pan-PPAR agonist TTA [28], and suggest that these biomarker alterations are due to PPARα activation.

In the PPARα group, higher plasma concentrations were observed for several metabolites of the choline oxidation pathway, including betaine, DMG, glycine and serine, with a majority of the observed variation in these variables ($\eta^2$ = 60–81%) being explained by the model. The

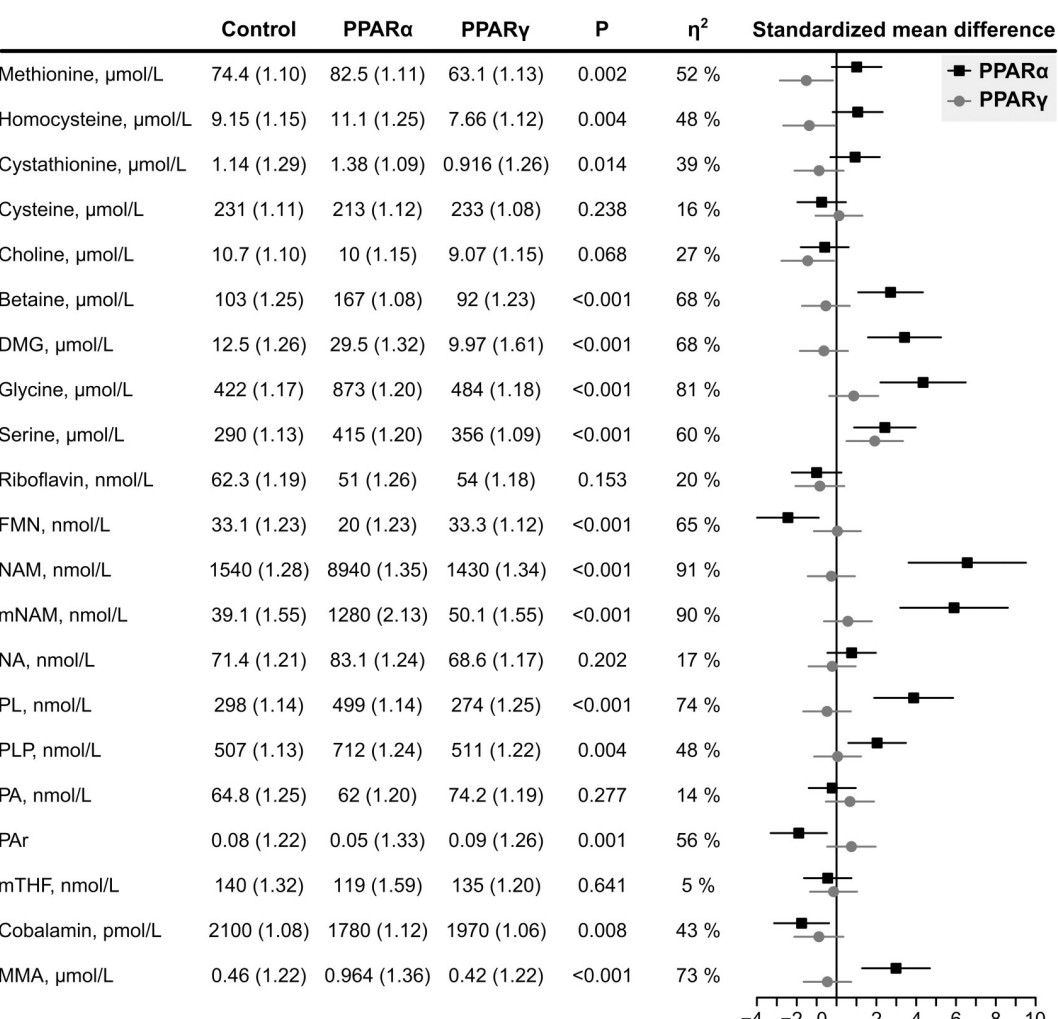

| | Control | PPARα | PPARγ | P | η² | Standardized mean difference |
|---|---|---|---|---|---|---|
| Methionine, µmol/L | 74.4 (1.10) | 82.5 (1.11) | 63.1 (1.13) | 0.002 | 52 % | |
| Homocysteine, µmol/L | 9.15 (1.15) | 11.1 (1.25) | 7.66 (1.12) | 0.004 | 48 % | |
| Cystathionine, µmol/L | 1.14 (1.29) | 1.38 (1.09) | 0.916 (1.26) | 0.014 | 39 % | |
| Cysteine, µmol/L | 231 (1.11) | 213 (1.12) | 233 (1.08) | 0.238 | 16 % | |
| Choline, µmol/L | 10.7 (1.10) | 10 (1.15) | 9.07 (1.15) | 0.068 | 27 % | |
| Betaine, µmol/L | 103 (1.25) | 167 (1.08) | 92 (1.23) | <0.001 | 68 % | |
| DMG, µmol/L | 12.5 (1.26) | 29.5 (1.32) | 9.97 (1.61) | <0.001 | 68 % | |
| Glycine, µmol/L | 422 (1.17) | 873 (1.20) | 484 (1.18) | <0.001 | 81 % | |
| Serine, µmol/L | 290 (1.13) | 415 (1.20) | 356 (1.09) | <0.001 | 60 % | |
| Riboflavin, nmol/L | 62.3 (1.19) | 51 (1.26) | 54 (1.18) | 0.153 | 20 % | |
| FMN, nmol/L | 33.1 (1.23) | 20 (1.23) | 33.3 (1.12) | <0.001 | 65 % | |
| NAM, nmol/L | 1540 (1.28) | 8940 (1.35) | 1430 (1.34) | <0.001 | 91 % | |
| mNAM, nmol/L | 39.1 (1.55) | 1280 (2.13) | 50.1 (1.55) | <0.001 | 90 % | |
| NA, nmol/L | 71.4 (1.21) | 83.1 (1.24) | 68.6 (1.17) | 0.202 | 17 % | |
| PL, nmol/L | 298 (1.14) | 499 (1.14) | 274 (1.25) | <0.001 | 74 % | |
| PLP, nmol/L | 507 (1.13) | 712 (1.24) | 511 (1.22) | 0.004 | 48 % | |
| PA, nmol/L | 64.8 (1.25) | 62 (1.20) | 74.2 (1.19) | 0.277 | 14 % | |
| PAr | 0.08 (1.22) | 0.05 (1.33) | 0.09 (1.26) | 0.001 | 56 % | |
| mTHF, nmol/L | 140 (1.32) | 119 (1.59) | 135 (1.20) | 0.641 | 5 % | |
| Cobalamin, pmol/L | 2100 (1.08) | 1780 (1.12) | 1970 (1.06) | 0.008 | 43 % | |
| MMA, µmol/L | 0.46 (1.22) | 0.964 (1.36) | 0.42 (1.22) | <0.001 | 73 % | |

**Fig 2. Biomarker concentrations, given as geometric mean (geometric standard deviations).** The groups were compared with one-way ANOVA, and the η2 indicates the proportion of variation explained by the model. The black lines with squares correspond to the standardized mean difference (95% CI) between the PPARα group and control, and the grey lines with circles represents PPARγ vs control. DMG indicates dimethylglycine; FMN, Flavin mononucleotide; MMA, methylmalonic acid; mNAM, N1-methylnicotinamide; mTHF, 5'-methyltetrahydrofolate; NA, nicotinic acid; NAM, nicotinamide; PA, 4-pyridoxic acid; PAr, PA/(PLP+PL); PL, pyridoxal; PLP, pyridoxal 5'-phosphate.

increased concentrations of these metabolites may indicate increased flux through the choline oxidation pathway. Although plasma mTHF did not differ between groups, the lower cobalamin and higher MMA levels observed in the PPARα group may indicate lower cobalamin availability. This could reduce the activity of the MS-mediated homocysteine remethylation pathway, causing a compensatory increase in BHMT flux. PPARα-induced down regulation of DMGDH and SARDH may cause accumulation of their upstream metabolites, DMG and sarcosine. Unfortunately, we were not able to determine plasma concentrations of sarcosine, due to analytical interference from the EDTA in the tubes used for blood sampling. Both DMGDH and SARDH are flavoproteins and lower concentrations of FMN may indicate lower cofactor availability, potentially contributing to reduced enzyme activities.

Compared to control, plasma glycine and serine concentrations were much higher in the PPARα group, which have been consistently found in other animal studies [18, 28–31, 43].

Ericsson et al demonstrated that the increased plasma glycine levels were primarily due to increased entry into plasma, possibly reflecting increased de novo synthesis [29]. In addition to the choline oxidation pathway, glycine may originate from carnitine synthesis, threonine catabolism or from glycolysis via serine [44, 45]. Increased carnitine synthesis has been demonstrated after PPARα activation [46], in line with the known increases in β-oxidation and lowering of triglycerides. Hence, increased carnitine synthesis following PPARα activation is a likely contributor to increased plasma glycine. However, circulating concentration of carnitines were lower after PPARα agonistic treatment in the current experiment [36]. PPARα is known to increase lipid oxidation and sparing glucose [47], hence, glucose metabolism is not a likely contributor to the observed increases in plasma glycine and serine in the PPARα group. Higher plasma serine may be related to interconversion of excess glycine through serine-hydroxymethyltransferase. Upstream accumulation of glycine and serine following PPARα-induced down-regulation of GNMT is another potentially contributing factor [18]. Higher plasma serine observed after treatment with PPARγ agonist may be related to increased glycolysis, as PPARγ activation is known to increase insulin sensitivity, facilitating increased glucose utilization, as well as directly influence hepatic glucose uptake [48].

As in our previous report we observe lower plasma levels of the vitamin B2 metabolite FMN after treatment with PPARα agonist [28]. This could reflect reduced formation of FMN from riboflavin through riboflavin kinase or increased metabolism of FMN to flavin adenine dinucleotide (FAD) by FAD synthase. Notably, FAD acts as a product inhibitor of its own synthesis [49]. Hence, increased FAD utilization may increase FAD synthesis, and thus reduce FMN and riboflavin levels. FAD acts as an electron acceptor in the first step of β-oxidation, and although speculative, this may suggest that PPARα induced increases in hepatic fatty acid oxidation may drive the conversion from FMN to FAD, by product removal. Lower plasma riboflavin and FMN may thus reflect the altered cofactor requirements following PPARα induced changes in energy metabolism.

The largest response to intervention in the current study were the higher concentrations of plasma NAM and mNAM in the PPARα group, with an excess of 90% of the observed variation being explained by the model. We [28] and others [18, 28, 31, 34, 35, 50, 51] have previously observed a marked increase in B3 vitamers in plasma, urine and liver of rodents treated with PPARα agonists. Thus, these observations fits well with the existing literature. In addition to dietary consumption, NAM may be endogenously formed from the amino acid tryptophan through the kynurenine pathway [52]. Amino-carboxymuconate semialdehyde (ACMS), a downstream metabolite of this pathway, is either catabolized by the enzyme ACMS dehydrogenase or converted to quinolinic acid and metabolized further to NAM through quinolinic acid phosphoribosyltransferase (QAPRT). PPARα activation has been shown to down regulate ACMS dehydrogenase and up regulate QAPRT [53, 54], shifting the flux towards increased niacin synthesis from tryptophan. We believe this offers a mechanistic explanation of our observations, suggesting NAM and/or mNAM as potential biomarkers of PPARα activity. This has to be examined in future studies.

Similar to our previous findings [28], we observe higher concentrations of both circulating B6 vitamers PL and PLP in the PPARα group. Vitamin B6 status is inversely associated with inflammation [55], and PLP has been suggested as a scavenger of reactive oxidative species [56]. Hence, the increased plasma concentrations of PL and PLP may be related to the anti-inflammatory and anti-oxidative properties of PPARα activation [17]. Of interest, we observe larger increases in PL compared to PLP. One possible explanation may be increased conversion from PLP to PL, a reaction catalyzed by alkaline phosphatase [56]. Increased alkaline phosphatase mRNA has been demonstrated after PPARα activation in cell culture studies [57]. Additionally, the final step in the synthesis of PLP from its precursors pyridoxine-5'-phosphate

and pyridoxamine-5'-phospate is catalyzed by pyridoxamine 5'-phosphate oxidase which is a FMN-dependent protein [56, 58], and lower FMN availability may contribute to reduced PLP synthesis via this route. This is also in line with observations that fibrate treatment does not change plasma PLP concentrations in humans [33].

We observed a large increase in plasma MMA after treatment with PPARα agonist, in accordance with what we previously reported after treatment with pan-PPAR agonist [28]. Although MMA is considered a functional biomarker of vitamin B12 deficiency, much of the variation in plasma MMA is not accounted for by vitamin B12 status, suggesting that B12 independent mechanisms might also be at play [59]. Of interest, Molloy *et al* identified a novel genetic variant in the metabolism of branched-chain amino acids (BCAA), explaining 9.9% of the variance of circulating MMA, independent of vitamin B12 status [60].

There are several possible ways PPARα activation may contribute to increased MMA; by increasing the availability of precursors, by interfering with intracellular vitamin B12 processing or by directly interfering with the methylmalonyl-CoA mutase reaction. The precursor of methylmalonyl-CoA is propionyl-CoA, resulting from catabolism of BCAA and odd-chained fatty acids. PPARα activation is well known to stimulate hepatic fatty acid oxidation [61] and has also been demonstrated to increase the catabolism of BCAA [62], suggesting an effect of PPARα activation on availability of MMA precursors.

A key step in the intracellular processing of cobalamin is the liberation of free cobalamin within the cell cytosol, a reaction involving by the MMACHC protein, and partitioning to the two cobalamin dependent proteins, facilitated by the MMADHC protein. MMACHC is a flavoprotein dependent on glutathione (GSH) for its function [63], and the availability of both GSH and FAD may be limited under conditions of increased PPARα activity. GSH production is directly related to the level of oxidative stress, and as much as half of the cysteine utilized for GSH production is derived from homocysteine through the transsulfuration pathway [64]. Anti-oxidative properties of PPARα agonists [17], combined with down regulation of both enzymes of the transsulfuration pathway [18, 19], may reduce GSH availability following PPARα activation. Further, *mmachc* and *mmadhc* are predicted PPAR target genes [65]. Hence, PPARα induced interference with intracellular processing of vitamin B12 and its availability as co-factor of methylmalonyl-CoA mutase, may contribute to increased MMA.

## Strengths and limitations

The main strength of this study is the use of specific PPARα and PPARγ agonists, allowing for evaluation of the specific effects of PPAR subtypes. As animals residing in the same cage belonged to the same experimental group, the possibility of cage effects could not be completely excluded, although the animals were individually treated outside the cages. However, by plotting cage mean values instead of individual measurements, the same patterns emerged and within-group cage means were similar (S2 Fig) to the results obtained by using the individual animals. Hence, bias due to cage effects is not likely to have influenced our main conclusions. It has to be noted that the effect of PPAR agonists on one-carbon and B-vitamin metabolome was a secondary aim of this experiment. However, maximizing the information gained from the animals contributes to reducing the number of animals sacrificed, which is in line with the 3 R's of animal research [66]. We did not measure PPARα and PPARγ activity or protein expression in this study, which may be considered a limitation. Measuring protein expression of PPARs would not necessarily add any information regarding their activity, since this involves several factors. However, activation of the PPARs with the agonists administered in this study has been thoroughly demonstrated in the literature, and increased expression of known PPARα and PPARγ target genes were observed in the current experiment [36].

Further, the discussion on potential mechanisms would benefit from information regarding the expression of genes along the pathways discussed. Unfortunately, as this information could not be obtained from the current experiment, the mechanistic discussion relies on effects previously reported in the literature. However, as previous publications have not provided complete information regarding effects on the metabolite level, the current data should be interpreted as complimentary to the existing literature.

## Conclusion

In this study, short-term treatment with a specific PPARα, but not PPARγ, agonist influenced plasma concentrations of several one-carbon metabolites and markers of B-vitamin status, with the most pronounced findings being higher DMG, glycine, serine, NAM, mNAM, PL and MMA, and lower FMN. This targeted metabolomics approach adds to the current literature suggesting the involvement of PPARα in the regulation of these metabolic pathways, as well as the status of closely related B-vitamins.

## Supporting information

**S1 Arrive Checklist.**
(PDF)

**S1 Fig. Individual metabolite concentrations with overlaying box plots.** Black dots represents the control group, red dots represents the PPARα group and blue dots represents the PPARγ group. DMG, dimethylglycine; FMN, Flavin mononucleotide; MMA, methylmalonic acid; mNAM, methylnicotinamide; mTHF, 5'-methyltetrahydrofolate; NA, nicotinic acid; NAM, nicotinamide; mNAM, N1-methylnicotinamide; PA, 4-pyridoxic acid; pABG, para-aminobenzoylglutamate; PAr, PA/(PLP+PL); PL, pyridoxal; PLP, pyridoxal 5'-phosphate. (TIF)

**S2 Fig. Cage mean metabolite concentrations with overlaying box plots.** Black dots represents the control group, red dots represents the PPARα group and blue dots represents the PPARγ group. DMG, dimethylglycine; FMN, Flavin mononucleotide; MMA, methylmalonic acid; mNAM, methylnicotinamide; mTHF, 5'-methyltetrahydrofolate; NA, nicotinic acid; NAM, nicotinamide; mNAM, N1-methylnicotinamide; PA, 4-pyridoxic acid; pABG, para-aminobenzoylglutamate; PAr, PA/(PLP+PL); PL, pyridoxal; PLP, pyridoxal 5'-phosphate. (TIF)

**S1 Dataset. Raw data on metabolite concentrations.**
(XLSX)

## Acknowledgments

The authors would like to thank Kari Mortensen, Kari Williams, Randi Sandvik, Liv Kristine Øysæd, Svein Kruger, Marte Trollebø and Torunn Eide for their valuable technical assistance during the study. We also thank the staff at the Laboratory Animal Facility at the University of Bergen.

## Author Contributions

**Conceptualization:** Bodil Bjørndal, Mari Lausund Grinna, Rolf Kristian Berge, Elin Strand.

**Data curation:** Vegard Lysne, Bodil Bjørndal, Mari Lausund Grinna, Øivind Midttun, Rolf Kristian Berge, Elin Strand.

**Formal analysis:** Vegard Lysne, Elin Strand.

**Investigation:** Mari Lausund Grinna, Per Magne Ueland, Jutta Dierkes, Ottar Nygård.

**Methodology:** Bodil Bjørndal, Øivind Midttun, Per Magne Ueland, Jutta Dierkes, Ottar Nygård.

**Project administration:** Vegard Lysne, Rolf Kristian Berge, Elin Strand.

**Visualization:** Vegard Lysne.

**Writing – original draft:** Vegard Lysne.

**Writing – review & editing:** Vegard Lysne, Bodil Bjørndal, Mari Lausund Grinna, Øivind Midttun, Per Magne Ueland, Rolf Kristian Berge, Jutta Dierkes, Ottar Nygård, Elin Strand.

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
