## [Decision Letter · Decision Letter 0]

8 Oct 2019

PONE-D-19-18869

Short-term treatment with a peroxisome proliferator-activated receptor α agonist influences plasma one-carbon metabolites and B-vitamin status in rats

PLOS ONE

Dear Mr. Lysne,

Thank you for submitting your manuscript to PLOS ONE. After careful consideration, we feel that it has merit but does not fully meet PLOS ONE’s publication criteria as it currently stands. Therefore, we invite you to submit a revised version of the manuscript that addresses the points raised during the review process.

The reviewer finds your study to be of interest; however they raise a number of issues which you should consider in revising your manuscript. In particular, please respond to the comment regarding hepatic PPARγ activation in your previous publication (ref 36) as it relates to the current study. The reviewer also requests that you perform some additional analyses to avoid duplication and build upon results presented in your previous publication.

We would appreciate receiving your revised manuscript by Nov 22 2019 11:59PM. To enhance the reproducibility of your results, we recommend that if applicable you deposit your laboratory protocols in protocols.io, where a protocol can be assigned its own identifier (DOI) such that it can be cited independently in the future. For instructions see: http://journals.plos.org/plosone/s/submission-guidelines#loc-laboratory-protocols

We look forward to receiving your revised manuscript.

Kind regards,

Nick Ashton, PhD

Academic Editor

PLOS ONE

**Journal Requirements:**

2. Please complete and submit a copy of the ARRIVE Guidelines checklist, a document that aims to improve experimental reporting and reproducibility of animal studies for purposes of post-publication data analysis and reproducibility: https://www.nc3rs.org.uk/arrive-guidelines. Please include your completed checklist as a Supporting Information file. Note that if your paper is accepted for publication, this checklist will be published as part of your article.

3. Please  review your funding statement as per our policy on disclosure of funding sources (http://journals.plos.org/plosone/s/disclosure-of-funding-sources) .

4. We note that one or more of the authors are employed by a commercial company: Bevital AS.

**Comments to the Author**

1. Is the manuscript technically sound, and do the data support the conclusions?

Reviewer #1: Partly

2. Has the statistical analysis been performed appropriately and rigorously? 

Reviewer #1: N/A

3. Have the authors made all data underlying the findings in their manuscript fully available?

Reviewer #1: Yes

4. Is the manuscript presented in an intelligible fashion and written in standard English?

Reviewer #1: Yes

5. Review Comments to the Author

Reviewer #1: The results showed in the manuscript give some significent explain of the effect of PPARα and PPARγ on one-carbon matbolism. However, the paper is unsuitable for publication in its present form. There are some problem and suggestion as followed.

1 Energy intake should be describle more detail.

2 Although the PPARα and PPARγ mRNA expression were anaylazied ( in reference 36 ), there no difference of PPARγ mRNA level between PPARγ and control, if PPARγ was not activated in haptic and adipose tissue, how do you get the conclusion that PPARγ activatoin has rarely accosiated with one carbon matebolism? In fact, according the experiment data showed in reference 36. It seemed that PPARα mRNA expression significantly increased compared with control only in heptaic tissue, howerer, it did not seem the PPARα agonist had significant effect on PPARα expression in adipose tissue.

3 It is not approved to separate the data of PPAR expression from the data of one-carbon matebolism products. Please provide data of wester-blotting or ELISA to certify PPARα and PPARγ expression at higher level under the intervention of agonist in current study, avoiding repetition data in literature 36.

4 The describe of PPARα expression should be showed in discussion in line 91-94.

5 The analysis of genes expresssion of serine-hydroxymethyltransferase, ACMS dehydrogenase , QAPRT, methylmalonyl-CoA mutase are necessary, otherwise, the avalible evidence is not very convincing.

6. PLOS authors have the option to publish the peer review history of their article (what does this mean?). If published, this will include your full peer review and any attached files.

Reviewer #1: No

---

## [Author Response · Author response to Decision Letter 0]

4 Nov 2019

1. Energy intake should be described more detail.

A: We have added a comment on how feed intake was estimated on line 92. Feed intake was higher in the PPARγ-group, and that group also gained slightly more total body weight as compared to the other groups. 

2. Although the PPARα and PPARγ mRNA expression were anaylazied ( in reference 36 ), there no difference of PPARγ mRNA level between PPARγ and control, if PPARγ was not activated in haptic and adipose tissue, how do you get the conclusion that PPARγ activatoin has rarely accosiated with one carbon matebolism? In fact, according the experiment data showed in reference 36. It seemed that PPARα mRNA expression significantly increased compared with control only in heptaic tissue, howerer, it did not seem the PPARα agonist had significant effect on PPARα expression in adipose tissue.

A: Activation of PPARs does not necessarily lead to increased PPAR mRNA, so the lack of effect on the gene level should not be interpreted as an indication of lack in activation. When the PPARs are activated, the genetic expression of their target genes are induced. In our previous publication, we demonstrate large increases in the hepatic expression of several PPARα target genes (e.g. Acox1, Cd36, LPL and Hmgcs2) and adipose tissue expression of PPARγ target genes (e.g. Fatp1 and Fabp4) in the respective treatment groups. As the PPAR-activating effect of the agonists used in this experiment are well established, demonstrating this was not a goal in the current experiment. However, we are very confident that the PPARs were activated, as demonstrated by the effect on target genes. 

To avoid confusion, we have replaced any reference to PPARα activation with treatment with PPARα-agonist, and PPARγ-activation with treatment with PPARγ-agonist, when discussing the present findings. 

3. It is not approved to separate the data of PPAR expression from the data of one-carbon matebolism products. Please provide data of wester-blotting or ELISA to certify PPARα and PPARγ expression at higher level under the intervention of agonist in current study, avoiding repetition data in literature 36.

A: As mentioned, data on PPAR expression would not inform any inference regarding PPAR activation. Also, due to lack of biological material, we are not able to perform additional laboratory analyses from this experiment. We hope the reviewer and editor accept that we are unfortunately not able to provide this data.

We do agree with the reviewer that some indices of PPAR activation should be included in the current manuscript. However, to avoid repetition from the main publication, we have included some results in the text with reference to the original publication (reference 36) on lines 95-99, as well as mentioning this as a limitation in the discussion section. 

4. The describe of PPARα expression should be showed in discussion in line 91-94.

A: We have added some data demonstrating PPARα and PPARγ-activation on lines 95-99, with reference to our previous publication.

5. The analysis of genes expresssion of serine-hydroxymethyltransferase, ACMS dehydrogenase , QAPRT, methylmalonyl-CoA mutase are necessary, otherwise, the avalible evidence is not very convincing.

A: We absolutely agree with the reviewer that this information would be very useful to aid the proposed mechanistic explanation. However, this was not included in the original protocol for the experiment, and due to lack of biological material it will unfortunately not be possible to perform additional laboratory analyses. 

However, we think our data still holds value, as we show that treatment with PPAR-agonists does influence the circulating metabolites. Although we, based on our data, cannot pinpoint the exact mechanisms, the pronounced effects on these metabolites cannot be ignored. We do think that concordance with previous literature showing an effect on the gene level strengthen our results, although we were not able to measure those enzymes in this specific experiment. We have added a sentence highlighting this limitation in the discussion section.

---

## [Editor Report · Decision Letter 1]

20 Nov 2019

Short-term treatment with a peroxisome proliferator-activated receptor α agonist influences plasma one-carbon metabolites and B-vitamin status in rats

PONE-D-19-18869R1

Dear Dr. Lysne,

We are pleased to inform you that your manuscript has been judged scientifically suitable for publication and will be formally accepted for publication once it complies with all outstanding technical requirements.

With kind regards,

Nick Ashton, PhD

Academic Editor

PLOS ONE
---

## [Editor Report · Acceptance letter]

25 Nov 2019

PONE-D-19-18869R1 

Short-term treatment with a peroxisome proliferator-activated receptor α agonist influences plasma one-carbon metabolites and B-vitamin status in rats 

Dear Dr. Lysne:

I am pleased to inform you that your manuscript has been deemed suitable for publication in PLOS ONE. Congratulations! Your manuscript is now with our production department. 

With kind regards,

on behalf of

Dr. Nick Ashton 

Academic Editor

PLOS ONE